# Biomechanic Differences Between Anticipated and Unanticipated Volleyball Block Jump: Implications for Lower Limb Injury Risk

**DOI:** 10.3390/life14111357

**Published:** 2024-10-23

**Authors:** Hongxin Zhao, Xiangyu Liu, Linfei Dan, Datao Xu, Jianshe Li

**Affiliations:** 1Department of Physical Education, Zhejiang Institute of Water Resources and Hydropower, Hangzhou 310000, China; 13252020777@163.com; 2Faculty of Sports Science, Ningbo University, Ningbo 315211, China; danlinfei@aliyun.com (L.D.); lijianshe@nbu.edu.cn (J.L.)

**Keywords:** team sports, lower limb injury, direction, unanticipated tasks, biomechanics

## Abstract

Volleyball is a high-intensity sport characterized by repetitive jumping, sudden directional changes, and overhead movements, all of which significantly increase the risk of injuries, particularly to the shoulders, knees, and ankles. Despite the frequency of injuries caused by actions like blocking and spiking, there has been limited research focused on the specific biomechanical risk factors unique to volleyball. This study aimed to investigate the lower limb biomechanics during block jumps in both the dominant and non-dominant directions, under both anticipated and unanticipated conditions, in fifteen elite male volleyball players. Kinematic and kinetic data from the ankle, knee, and hip joints were recorded. Our results revealed statistically significant differences between the dominant and non-dominant directions at the ankle, knee, and hip joints. The non-dominant direction exhibited a greater ankle dorsiflexion angle and velocity, as well as higher knee flexion angle, velocity, moment, power, and abduction moment, along with increased hip flexion angle and power. Additionally, unanticipated movements led to increases in vertical ground reaction force (vGRF), hip extension moment, and flexion power, while ankle dorsiflexion plantarflexion velocity and knee flexion power decreased. It appears that movements in the dominant direction were stiffer and less cushioned, potentially increasing the risk of injury. While the non-dominant direction provided better shock absorption, it also elevated the knee valgus moment, which could increase the load on the knee. Furthermore, in unanticipated situations, athletes with short reaction times, unable to quickly adjust their automated movement patterns, faced a higher risk of limb overuse, thereby increasing the likelihood of injury. In practice, coaches should consider differences in limb coordination and movement direction, incorporating unilateral preventive exercises to reduce the risk of injury.

## 1. Introduction

Athletes are affected by complex internal and external factors when performing specific sports tasks, which may be related to the risk of sports injuries [1,2,3]. Modern volleyball is characterized by its fast-paced nature, multiple playing levels, and rapid transitions between offense and defense [4,5,6]. Specific actions such as swift movements [7], smashing [8], and blocking, all put higher requirements on the volleyball player’s musculoskeletal system [2,3,9,10,11]. The complex internal and external factors combined with the potential dangers of volleyball-specific actions make volleyball one of the team events with the highest damage rate [7]. Epidemiological studies indicate that approximately 90% of injuries in volleyball occur in the lower limbs [12]. In particular, the incidence of knee injuries, including non-contact ACL ruptures, osteoarthritis, articular cartilage damage, and patellar tendinitis, is high. For instance, epidemiological studies have reported that knee injuries account for up to 30% of all volleyball-related injuries, with non-contact ACL injuries comprising a significant portion of these cases. The recovery process is not only lengthy but also financially burdensome, often requiring extended rehabilitation periods, surgery, and long-term management strategies [13,14,15]. Therefore, increasing the research on risk factors related to knee injury in volleyball is very important.

In volleyball, sports injuries may occur in any stage of fast movement, emergency stop, take-off, and landing in the blocking [6,9,16,17]. Unfortunately, many studies have predominantly focused on the biomechanical changes and risk factors for knee injuries during landing tasks, as the landing phase is known for high-impact forces and abrupt deceleration, increasing the likelihood of non-contact injuries like ACL tears [18,19,20]. However, the take-off phase, characterized by rapid power generation and intense muscle activation, also presents significant injury risks. Despite the biomechanical demands of the take-off—such as the sharp increase in quadriceps force and joint torque—only a few studies have investigated its role in contributing to knee and lower limb injuries. This lack of attention leaves a critical gap in understanding the full spectrum of risk factors across the entire volleyball movement cycle [21,22,23]. Volleyball block jump (VBJ) is a typical stop-jump task, which includes the fastest movement, emergency stop, and take-off [11]. Previous studies have indicated that the range of motion during stop-jumping exceeds that of landing [22,24,25]. Due to the requirement of fast movement, athletes often do not have enough time to fully flex the knee to cushion the ground reaction force [26]. Moreover, VBJ requires both abrupt changes in body direction and maximum take-off speed, which hinders athletes’ ability to promptly adjust their trunk posture and knee abduction angle to execute the required movement. When performing a block jumping task, rapid steering will cause the knee valgus to collapse, and the knee will change the original trajectory to move toward the inside of the limb, causing the distal angle of the calf to move away from the body’s midline of the body. Coupled with the large knee extension moment produced during take-off, VBJ has become a typical dangerous sport mode reflecting the risk of injury of volleyball players.

In volleyball matches, multi-point three-dimensional offensive tactics are commonly employed [27]. Each round of offensive positions has at least four attackers. This requires the blockers to change their movement patterns at any time according to their predictions and the position of the last ball passing. Unanticipated or unplanned situations will frequently occur during the game, and athletes must respond quickly to sudden changes [28]. Unfortunately, most studies have focused solely on the correlation between established sports patterns and specific injury risks, neglecting the use of unanticipated experimental models to simulate real game scenarios [6]. While independent interventions in anticipated environments aim to enhance sports control strategies and reduce injury incidence, they present certain limitations [10]. Compared to the anticipated environment, the unanticipated environment is more prone to sports injuries [29]. Recent studies have suggested that compared with the anticipated situation, the unanticipated task produces a greater posterior ground reaction force and has a shorter time to reach the peak of vertical ground reaction force [30]. Furthermore, the peaks of knee valgus and internal rotation angles are more pronounced during unanticipated tasks, suggesting that unanticipated factors elevate injury risks during specific movements [6,31]. The above results may be related to the theory of motion control. Under the anticipated circumstances, the subjects have sufficient preparation time, and the central nervous system will actively adjust the body posture so that the whole body is in a suitable starting posture, so there is no need to change the original sports pattern [26]. Under unanticipated circumstances, the subject will have a predictive effect on the action to be performed, and when unplanned instructions are issued, they will make quick posture adjustments according to the instructions [32]. In the rapid response mode, the body will reflexively adjust its posture and use the foot valgus landing and the tilt of the torso to complete the stop-jump task in VBJ [33].

Previous studies highlight lateral directional movements as critical factors in assessing limb injury risks [34]. When a volleyball player is trying to get the greatest block performance, they use a natural sequence of a three-step technique during the VBJ, which is determined by the dominant leg [10]. For example, for a person whose dominant limb is the right leg, his preferred movement pattern in the block should be right-left-right to move to the left area. This lateral directional movement is called the dominant direction (DD). Conversely, when moving to the right, their usual sequence would be left-right-left, identified as the non-dominant direction (NDD). However, athletes must choose the blocking direction of lateral movement according to the position of the opponent’s attack in volleyball competitions, which will change the three-step movement pattern they are used to, and thus change the movement pattern of the VBJ [35]. Therefore, exploring balanced motor patterns can help reduce the risk of injury through early prevention.

Given these considerations, utilizing an unanticipated test model that closely simulates real game environments is vital for identifying risk factors associated with lower extremity injuries during specific movements. Therefore, this research aims to comprehensively analyze the mechanisms underlying lateral movement in both dominant and non-dominant volleyball blocks under anticipated and unanticipated VBJ conditions. We hypothesized there would be different strategies between the dominant and non-dominant directions in sagittal planes. Furthermore, we hypothesized that there would be differences between anticipated and unanticipated situations in VBJ.

## 2. Materials and Methods

### 2.1. Study Design

The independent variables in this study comprised two main factors: (1) the type of situation, categorized into: (a) anticipated block jump and (b) unanticipated block jump; and (2) the direction of lateral movement, which included: (a) dominant direction and (b) non-dominant direction. The dominant limb was defined as the preferred leg for tasks such as kicking a ball. In this study, all participants exhibited right-leg dominance, identified as their preferred leg for power generation during take-off and directional movements in blocking. This assessment of lower limb dominance is essential for understanding movement efficiency and injury risk, as the dominant leg bears a greater load during explosive actions and stabilizes the body in critical phases of play [6,36].

In this research, we considered anticipated and unanticipated situations before the start of the volleyball block approach. To ensure the integrity of the results, we implemented several strategies to control potential confounding variables. Fatigue was minimized through adequate rest periods between trials and a standardized warm-up routine, while practice trials were incorporated to mitigate learning effects, allowing participants to familiarize themselves with the tasks and equipment. These strategies created a more reliable testing environment. In this context, anticipated means that the subject has sufficient time and self-consciousness to choose the direction of lateral movement when performing the blocking task. The term ‘unanticipated’ refers to the immediate initiation of the blocking approach upon the illumination of one of the three lights, requiring participants to react without the opportunity for conscious planning. The light cues were randomized using a computer algorithm to ensure unpredictability in their sequence. In both the anticipated and unanticipated conditions, participants were instructed to reach the designated location as quickly as possible. The anticipated condition mirrored typical training scenarios, such as daily ball-free blocking drills or situations with a single attacker. However, unanticipated situations correspond to scenarios where there are multiple attackers in a volleyball game. In such cases, players must contend with three possible attacks that are displayed randomly, and their task is to move and block these attacks in the shortest possible time. This defensive strategy is not only frequently used by middle blockers; the outside blockers on both sides also face the same complex defensive situation, so it is difficult for them to defend against all possible attack positions (Figure 1). The volleyball block jumping biomechanical characteristics were analyzed to see whether there were differences in movement strategies under the influence of anticipated and unanticipated factors during VBJ.

### 2.2. Participants

This study involved 15 male volleyball players with an average age of 20.83 years (±1.34), height of 190.8 cm (±3.57), and body mass of 80.9 kg (±3.98). The participants were selected from a local university, where they competed in the National College Volleyball League. Each player was recognized as a national second-level athlete, training consistently five times a week. All subjects received comprehensive information about the study, including potential risks. Informed consent was obtained, adhering to the approval requirements from the Institutional Review Board at Ningbo University, China. The eligible criteria for subjects were as follows: (1) no history of significant musculoskeletal injuries, including ligament tears, fractures, tendon injuries, or chronic joint pain; (2) no prior surgical procedures related to the lower limbs or spine; (3) no new musculoskeletal injuries within the past year; (4) for participants with injuries within the past two years, documentation or self-report of injury severity must be provided; (5) a minimum of two years of volleyball playing experience to ensure familiarity with the sport; (6) no neurological disorders that could impact motor skills or coordination [6].

### 2.3. Experimental Setup

In this study, a simulated volleyball net was constructed in the laboratory to replicate real game conditions. To capture kinematic data, we employed an infrared high-speed motion analysis system featuring eight cameras (Vicon, Oxford Metrics Ltd., Oxford, UK, 200 Hz) arranged around the court. For kinetic measurements, a force platform (AMTI, Watertown, MA, USA, 1000 Hz) was installed at the court’s center. Both systems were synchronized to ensure accurate data collection during the movement trials (refer to Figure 2). We utilized a set of 20 reflective markers, each with a diameter of 12.5 mm, for tracking the participants’ movements. To construct an unanticipated experimental model, the subjects performed a flight Trainer sequence programming protocol. In the formal experiment, a light is used as the visual target, which indicates the moving direction of the block, and this was also used to detect whether the VBJ height qualified (Figure 3).

### 2.4. Protocol

The experiment was set in a laboratory environment that simulated real games, and the height of the volleyball net was 2.43 m. To standardize the jump height during unanticipated scenarios, three light discs were suspended 0.40 m above the top of the net on the opponent’s side of the court, simulating an attacking move to evaluate block effectiveness. In both anticipated and unanticipated situations, participants were instructed to reach the designated position as quickly as possible. However, in anticipated scenarios, they had the flexibility to start at their discretion without time constraints, which allowed for conscious planning of their movements. In unanticipated situations, the subject moved to the designated area as quickly as possible, and without preparation time, according to the direction of the position where the three lights were randomly lit. In addition, the evaluator assessed if dominant limbs fell on the force platform, but during the experiment, the subject was reminded to try not to pay attention to the position of the force plate, so as to not affect the experimental actions.

Participants underwent a 20 min warm-up of light jogging and static stretching before completing five familiarization trials in the testing area. They performed the VBJ from randomized directions (left or right). A 5 min rest period was provided between trials to minimize fatigue. Then, the protocol was repeated in the opposite direction. After each sequence, the Borg scale was used to assess fatigue and control it, keeping it below the threshold of fifteen. There was a coach on-site to supervise the experiment in the test to ensure a qualified trial. The accepted trials had to meet the following criteria: (1) a natural and powerful trial as judged by the coach, (2) a frontal jump, (3) a short lateral jump, (4) a three-step block approach, and (5) the dominant foot rests completely on the force plate. All experiments that did not meet any of the above criteria were discarded. For each subject, at least five trials were collected under each situation and each direction [36].

### 2.5. Data and Statistical Analysis

The raw data for marker positions and ground reaction forces were processed using a fourth-order Butterworth low-pass filter, set at 7 Hz for marker data and 15 Hz for ground reaction forces [37]. A threshold of 10 N for vertical ground reaction force (vGRF) was used to identify the initial and take-off points of the VBJ [38]. Kinetic and kinematic analysis of the ankle, knee, and hip was performed using a hybrid rigid lower limb model in Visual 3D (C-Motion, Bethesda, MD, USA). Jump height was computed based on the center of gravity, with trials evaluated within 5% of the maximum recorded jump height. Movement consistency was assessed by examining the approach velocity on the force plate. The anterior shear force on the proximal tibia encompasses both soft tissue and joint contact forces at the knee, indicating the shear force applied to the ACL and transmitted by the patellar tendon [2,39,40].

All discrete variables were confirmed to have a normal distribution through the Shapiro–Wilk test. We applied a 2 × 2 repeated measures ANOVA to evaluate differences between dominant and non-dominant directions, as well as between anticipated and unanticipated tasks. Post hoc analyses were conducted with Bonferroni corrections to mitigate type I error, using a significance level of 0.05. Statistical analyses were performed with IBM SPSS Statistics 24.0 (SPSS, Inc., Chicago, IL, USA).

## 3. Results

### 3.1. Kinematics

The kinematic variables for the dominant ankle, knee, and hip joints during VBJ are shown in Table 1. There were significant differences in the angles of the ankle, knee, and hip between the dominant and non-dominant directions, with the non-dominant directions showing greater maximal dorsiflexion angle (F (1,13) = 31.192, *p* < 0.001, ƞ^2^ = 0.706) and maximal knee flexion angle (F (1,13) = 12.474, *p* = 0.004, ƞ^2^ = 0.490) and a greater flexion in the hip (F (1,13) = 8.437, *p* = 0.012, ƞ^2^ = 0.394).

A significantly greater peak ankle dorsiflexion (F (1,13) = 235.241, *p* < 0.001, ƞ^2^ = 0.948) and peak knee flexion velocity (F (1,20) = 164.99, *p* < 0.001, ƞ^2^ = 0.892) were seen in the movements in the non-dominant direction, with a greater amount of knee flexion angle at initial contact (F (1,13) = 169.881, *p* < 0.001, ƞ^2^ = 0.929), and significant greater peak ankle plantarflexion (F (1,13) = 590.404, *p* < 0.001, ƞ^2^ = 0.978), peak knee extension velocity (F (1,13) = 28.127, *p* < 0.001, ƞ^2^ = 0.684), and peak hip flexion velocity (F (1,12) = 289.826, *p* < 0.001, ƞ^2^ = 0.960) when moving in the dominant direction.

For peak ankle dorsiflexion velocity and peak ankle plantarflexion velocity, differences were noted between anticipated and unanticipated tasks (F (1,13) = 17.739, *p* = 0.001, ƞ^2^ = 0.577 and F (1,13) = 8.148, *p* = 0.014, ƞ^2^ = 0.385). A statistically significant interaction was observed for the peak of ankle plantarflexion velocity (F (1,13) = 6.336, *p* = 0.026, ƞ^2^ = 0.328), and further analysis showed a statistically greater peak ankle plantarflexion velocity (F (1,13) = 8.157, *p* = 0.013, ƞ^2^ = 0.386) in the anticipated movements in the dominant direction (Figure 4). However, the peak of ankle plantarflexion velocity increased with both anticipated and unanticipated movements in the dominant direction (F (1,25) = 238.788, *p* < 0.001, ƞ^2^ = 0.905 and F (1,13) = 128.591, *p* < 0.001, ƞ^2^ = 0.908) (Table 1).

### 3.2. Kinetics

As shown in Table 2, a statistically significant interaction was observed for the peak of vertical ground reaction force (F (1,21) = 6.351, *p* = 0.020, ƞ^2^ = 0.232), showing a lower peak posterior ground reaction force in the non-dominant directions (F (1,16) = 212.941, *p* < 0.001, ƞ^2^ = 0.930). Further analysis revealed a significantly greater peak posterior ground reaction force (pGRF) during unanticipated movements in the dominant direction (F (1,21) = 8.783, *p* = 0.007, ƞ^2^ = 0.295) (Figure 4 and Table 2).

The kinetic variables for the dominant ankle, knee, and hip joints during the block jumping are shown in Table 3. Significant differences were observed in the dominant knee and hip between the dominant and non-dominant directions, with the non-dominant direction exhibiting greater peak knee flexion (F (1,12) = 14.522, *p* = 0.002, ƞ^2^ = 0.548) and abduction moment (F (1,12) = 19.301, *p* = 0.001, ƞ^2^ = 0.617), peak of knee flexion power (F (1,12) = 23.927, *p* < 0.001, ƞ^2^ = 0.666), and peak hip flexion power (F (1,13) = 14.784, *p* = 0.002, ƞ^2^ = 0.532). In contrast, the peak knee (F (1,12) = 11.936, *p* = 0.005, ƞ² = 0.499) and hip extension power (F (1,13) = 31.915, *p* < 0.001, ƞ² = 0.711) were found to decrease in the non-dominant direction, while the peak knee flexion power increased during unanticipated movements. A statistically significant interaction was observed for the peak hip extension moment (F (1,13) = 6.164, *p* = 0.027, ƞ^2^ = 0.322) and peak hip flexion power (F (1,13) = 14.784, *p* = 0.002, ƞ^2^ = 0.532), showing greater moment (F (1,13) = 6.763, *p* = 0.022, ƞ^2^ = 0.342) and power (F (1,13) = 6.574, *p* = 0.024, ƞ^2^ = 0.336) between anticipated and unanticipated for the non-dominant direction. Further analysis revealed statistically greater peak hip flexion power in the non-dominant direction for both anticipated (F (1,22) = 24.859, *p* < 0.001, ƞ² = 0.531) and unanticipated (F (1,13) = 14.575, *p* = 0.002, ƞ² = 0.529) movements (Table 3).

## 4. Discussion

The purpose of the present study was to compare lower extremity movements between the DD and NDD in anticipated and unanticipated situations during the VBJ in male athletes. The results of this study suggest that there were different strategies between the dominant directions and non-dominant directions when athletes performed a volleyball block jumping task. Furthermore, unanticipated situations may affect the musculoskeletal system more than anticipated conditions. This highlights the importance of considering not only limb dominance during lateral movements but also the need to simulate competition scenarios as closely as possible in daily training.

Previous studies in lower limb symmetry during take-off and landing missions have been controversial [41,42]. Some authors report that limb dominance was an important factor leading to lower limb injury [41], while others report that there were no differences between limbs, but that the movement sequence of limbs was the key factor leading to the change of lower limb movement strategies [10]. In agreement with Zahradnik et al. [11], we observed a similar response in the knee joint angle at initial contact, which is small and is not conducive to landing cushioning during the non-dominant direction task. However, as the movements progress, the hip and knee joints achieve greater flexion angles, which may reduce power absorption in these joints while enhancing cushioning. These biomechanical differences in movement patterns can be attributed to the specific demands placed on each limb during the block jump. In the non-dominant direction, the increased flexion at the hip and knee is crucial for absorbing impact forces, effectively mitigating injury risk. In contrast, the dominant direction maintains a more rigid posture, which is essential for sustaining forward momentum. Importantly, our findings align with prior studies that the non-dominant direction results in a heightened peak valgus moment at the knee. This is particularly significant, as numerous studies have established that larger peak valgus moments are associated with an increased risk of knee injuries [43]. The above results can be explained by the differences in the movements of different blocking directions. When moving in the non-dominant direction, the dominant limb located on the outside mainly plays a cushioning role due to the difference in the order of limb movement [17]. On the contrary, when moving in the dominant direction, the dominant limb located on the inside plays the role of braking and stirring [44]. Therefore, the non-dominant direction of the hip and knee are more flexed to play a cushioning role, but with the hip and knee flexion, valgus moments may increase, while the dominant direction is relatively more rigid in the cushioned position due to the need for speed. The increase in angular velocity of hip and knee extension also facilitates the process of jumping by providing momentum to the stirrup takeoff phase [45]. This rapid extension can lead to more forceful landings, imposing significant mechanical stress on the knee joint. The elevated pGRF observed in the dominant direction indicates that the dominant limb bears greater loads upon landing. Such heightened forces are critical in the mechanisms underlying anterior cruciate ligament (ACL) injuries, as they may induce excessive knee valgus and anterior shear forces on the tibia, thereby increasing the risk of ligament strain or rupture [46,47]. Furthermore, the dynamic nature of these movements is associated with an elevated risk of overuse injuries, such as patellar tendinitis, particularly when the knee is subjected to repetitive high forces without adequate recovery [48].

A recent report confirmed that when there is an unanticipated situation during stop-jumping, the automated processes of the neuromuscular system are disrupted and the original motor strategy is altered [49,50]. This shift can be understood through neural control theories, which describe how under anticipated conditions, top-down control mechanisms are employed, allowing athletes to consciously regulate movements. However, in unanticipated scenarios, the neuromuscular system switches to automatic processes that prioritize immediate goal achievement over sensory feedback, as explained by attention resource theory. The rapid reallocation of neural resources under these conditions can lead to excessive muscular loading, increasing the risk of injury due to a diminished capacity to process proprioceptive signals that would normally adjust movements to prevent overload [51]. The comparison between anticipated and unanticipated volleyball block jumps revealed that unanticipated jumps resulted in higher peak vertical ground reaction force (vGRF), along with greater hip extension moment and flexion power, indicating an increased risk of overuse injuries. In contrast, unanticipated jumps showed a reduction in knee flexion power and ankle dorsiflexion and plantarflexion velocities. This discrepancy may be attributed to the limited time available for proper cushioning and extension during unanticipated jumps, leading to instability in the ankle joint, which in turn affected its dorsiflexion and plantarflexion movements [52,53,54]. According to Wulf et al. [55], when performers use an internal focus of attention, this may limit original automatic control processes and enhance active control capabilities. According to the top-down central nervous system theory, when a goal orientation appears, the organism actively activates the discriminatory function of the relevant information [56]. The most plausible explanation is that intra-anticipatory movements reinforce the internal focus enhancing active control, while unanticipated situations focus more on automated processes of the body’s neuromuscular system [7].

In this study, we have created an experimental protocol that simulates a real race environment and integrated lower limb variables in the sagittal plane that have been previously reported as risk factors in lower limb injuries. Additionally, we strictly standardized the approach speed and distance to enhance the ecological validity of the variable differences observed. Despite its contributions, this study has several limitations. The relatively small sample size, focus on male athletes with uniform blocking techniques, and consideration of only lower limb movements may limit the generalizability of the findings [57]. Additionally, although participants were instructed to move as fast as possible and land with the dominant limb on the force platform, this controlled setup does not fully mirror the complexity of real game scenarios. A more significant limitation is the absence of muscle activation data, which could provide a deeper understanding of the neuromuscular strategies involved in volleyball block jumping. Future research should aim to simulate more realistic game conditions, incorporating varied stimuli and muscle activation analysis to better understand motor strategies in block jumping [28,58].

## 5. Conclusions

In conclusion, the analysis revealed different strategies employed by participants when performing block jumps in anticipated versus unanticipated situations. Our results suggest that movements in the dominant direction exhibited greater stiffness and less cushioning, thereby increasing the risk of injury. In contrast, while the non-dominant direction provided effective shock absorption, it also elevated the knee valgus moment, resulting in increased knee load. Furthermore, in unanticipated situations, for athletes with short reaction times to the point of not being able to make immediate adjustments to automated movement processes, this may lead to increased limb overuse, creating a greater risk of injury. In practical applications, coaches should consider the differences in limb coordination and movement directions by incorporating unilateral preventive exercises. Moreover, simulating competitive scenarios during training can better prepare athletes for unanticipated situations, fostering quicker decision-making and adaptive movement strategies. By enhancing athletes’ responsiveness and movement efficiency in unpredictable contexts, these training methods may significantly reduce the risk of injuries associated with excessive muscular loading.

## Figures and Tables

**Figure 1 life-14-01357-f001:**
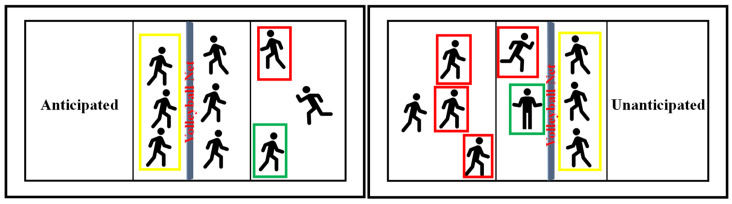
The red box is the attacker, the green box is the second passer, and the yellow box is the blocker. The anticipated situation can be interpreted as follows: the blocker can determine where the attacker is spiking the ball from, as in the left picture where there is only one attacker. In the unanticipated case, the blocker cannot tell who is going to attack, as, in the right picture, three blockers must face four attackers.

**Figure 2 life-14-01357-f002:**
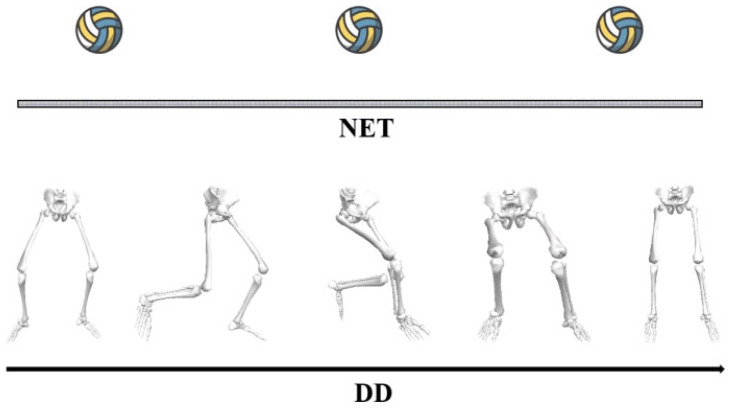
Volleyball blocking jumping in dominant direction.

**Figure 3 life-14-01357-f003:**
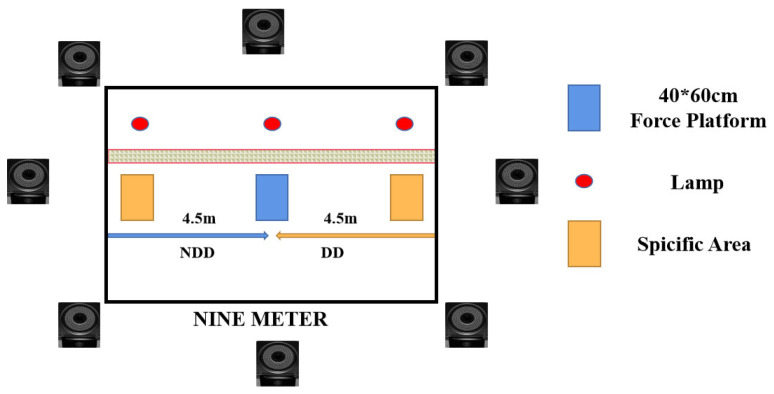
Top view of the experimental environment.

**Figure 4 life-14-01357-f004:**
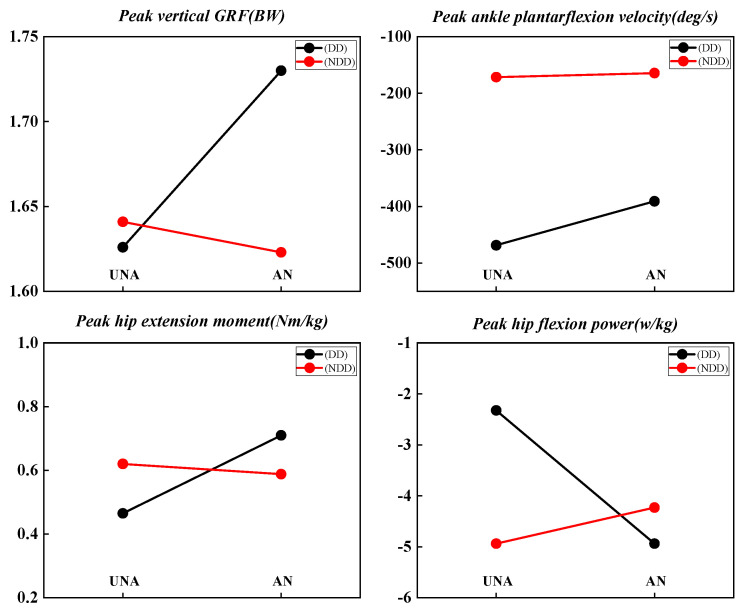
Visual representation of interactions between anticipated (AN)/unanticipated (UnA) and dominant direction (DD)/non-dominant direction (NDD) for peak vertical GRF (*p* = 0.020), peak ankle plantar-flexion velocity (*p* = 0.026), peak hip extension moment (*p* = 0.027), and peak hip flexion power (*p* = 0.002).

**Table 1 life-14-01357-t001:** Kinematic variables for the dominant limb joints during block jumping (mean ± standard deviation).

	Anticipated		Unanticipated			ANOVA *p*-Value			
	DD (SD)	NDD (SD)	DD (SD)	NDD (SD)	*p*-Value A v UnA	Effect Size	*p*-Value D	Effect Size	Interaction
Jump height (m)	0.58 (0.09)	0.57 (0.10)	0.57 (0.09)	0.60 (0.08)	0.444	0.020	0.618	0.009	0.240
Approaching velocity (m/s)	0.38 (0.05)	0.39 (0.04)	0.41 (0.12)	0.40 (0.09)	0.562	0.011	0.715	0.007	0.362
Ankle									
Ankle angle at contact (deg)	8.17 (5.42)	7.33 (6.85)	6.17 (6.72)	6.12 (6.57)	0.110	0.099	0.696	0.006	0.803
Peak ankle plantar-flexion angle (deg)	26.83 (5.62)	26.24 (6.77)	27.89 (7.26)	29.27 (7.04)	0.413	0.052	0.780	0.006	0.450
Peak ankle dorsiflexion angle (deg)	26.37 (6.22)	34.61 (4.65)	29.80 (2.76)	32.54 (5.28)	0.643	0.017	0.001 *	0.706	0.081
Peak ankle dorsiflexion velocity (deg/s)	121.37 (36.71)	344.13 (71.01)	109.64 (44.95)	251.63 (60.12)	0.001 *	0.577	0.000 *	0.948	0.055
Peak ankle plantar-flexion velocity (deg/s)	468.62 (40.45)	171.54 (29.66)	390.75 (83.31)	164.34 (29.85)	0.014 *	0.385	0.000 *	0.978	0.026 *
Knee									
Knee angle at contact (deg)	64.71 (16.17)	40.33 (9.45)	70.25 (8.06)	42.62 (12.11)	0.454	0.044	0.000 *	0.929	0.417
Peak knee flexion angle (deg)	81.60 (16.00)	86.72 (4.36)	79.66 (13.00)	94.79 (13.61)	0.517	0.033	0.004 *	0.490	0.126
Peak knee flexion velocity (deg/s)	89.86 (46.98)	356.92 (107.84)	78.12 (57.02)	367.04 (116.05)	0.965	0.000	0.000 *	0.892	0.611
Peak knee extension velocity (deg/s)	437.83 (104.76)	306.17 (45.59)	309.64 (156.81)	284.11 (31.70)	0.063	0.242	0.000 *	0.684	0.086
Peak knee abduction angle (deg)	2.56 (4.39)	1.34 (3.42)	2.54 (4.95)	2.54 (4.95)	0.769	0.007	0.330	0.079	0.452
Hip									
Peak hip flexion angle (deg)	57.59 (15.94)	67.26 (18.22)	58.53 (13.56)	66.90 (16.14)	0.959	0.000	0.012 *	0.394	0.801
Peak hip flexion velocity (deg/s)	213.76 (54.49)	85.18 (14.39)	185.74 (40.83)	72.50 (22.70)	0.189	0.139	0.000 *	0.960	0.491

* Significance (*p* < 0.05); significant interaction between non-dominant/dominant direction and anticipated (A)/unanticipated (UnA). SD—standard deviation; Deg—degrees; N—Newton; m—meter; kg—kilogram; M—moment joint; ω—angular velocity; J—Joule; s—second.

**Table 2 life-14-01357-t002:** Kinetic variables for the dominant limb during block jumping (mean ± standard deviation).

	Anticipated		Unanticipated			ANOVA *p*-Value			
	DD (SD)	NDD (SD)	DD (SD)	NDD (SD)	*p*-Value A v UnA	Effect Size	*p*-Value D	Effect Size	Interation
Peak posterior GRF (BW)	0.67 (0.08)	0.39 (0.08)	0.67 (0.08)	0.41 (0.10)	0.632	0.015	0.001 *	0.930	0.561
Peak vertical GRF (BW)	1.63 (0.18)	1.64 (0.20)	1.73 (0.19)	1.62 (0.12)	0.126	0.108	0.386	0.036	0.020 *
Mean loading rate to the vGRF (BW/s)	41.59 (11.82)	39.09 (14.14)	54.37 (11.94)	53.86 (10.30)	0.067	0.150	0.875	0.001	0.891

* Significance (*p* < 0.05=); significant interaction between non-dominant/dominant direction and anticipated (A)/unanticipated (UnA). SD—standard deviation; Deg—degrees; N—Newton; m—meter; kg—kilogram; M—moment joint; ω—angular velocity; J—Joule; s—second.

**Table 3 life-14-01357-t003:** Kinetic variables for the dominant limb joints during a block jumping (mean ± standard deviation).

	Anticipated		Unanticipated			ANOVA *p*-Value			
	DD (SD)	NDD (SD)	DD (SD)	NDD (SD)	*p*-Value A v UnA	Effect Size	*p*-Value D	Effect Size	Interaction
Ankle									
Peak ankle plantar-flexion moment (Nm/kg)	0.02 (0.05)	0.04 (0.06)	0.02 (0.02)	0.04 (0.05)	0.725	0.011	0.251	0.108	0.982
Peak ankle dorsiflexion moment (Nm/kg)	−2.25 (0.41)	2.12 (0.32)	2.33 (0.50)	2.12 (0.31)	0.725	0.011	0.069	0.250	0.662
Peak ankle plantar-flexion power (w/kg)	16.10 (2.75)	16.71 (3.34)	16.99 (3.65)	17.49 (3.09)	0.528	0.031	0.405	0.054	0.933
Peak ankle dorsiflexion power (w/kg)	3.13 (1.24)	3.61 (0.73)	4.02 (1.71)	3.34 (0.72)	0.389	0.058	0.719	0.010	0.097
Knee									
Peak knee flexion moment (Nm/kg)	0.31 (0.19)	0.50 (0.31)	0.25 (0.18)	0.39 (0.20)	0.421	0.055	0.002 *	0.548	0.456
Peak knee extension moment (Nm/kg)	2.85 (0.22)	2.94 (0.32)	2.81 (0.26)	2.96 (0.29)	0.908	0.001	0.120	0.189	0.632
Peak knee abduction moment (Nm/kg)	0.12 (0.18)	0.54 (0.27)	0.15 (0.10)	0.40 (0.38)	0.077	0.238	0.001 *	0.617	0.629
Peak knee flexion power(w/kg)	6.68 (1.49)	9.88 (2.09)	5.47 (2.39)	9.08 (1.90)	0.027 *	0.346	0.000 *	0.666	0.734
Peak knee extension power(w/kg)	18.35 (1.92)	15.94 (1.35)	19.21 (2.24)	17.32 (2.21)	0.081	0.232	0.005 *	0.499	0.671
Proximal anterior tibia shear force (BW)	0.55 (0.31)	0.47 (0.28)	0.43 (0.26)	0.41 (0.18)	0.194	0.117	0.292	0.079	0.606
Hip									
Peak hip flexion moment (Nm/kg)	2.61 (0.64)	2.88 (1.06)	2.74 (1.04)	3.00 (0.86)	0.565	0.028	0.202	0.132	0.992
Peak hip extension moment (Nm/kg)	0.46 (0.23)	0.62 (0.25)	0.71 (0.34)	0.59 (0.18)	0.083	0.214	0.847	0.003	0.027 *
Peak hip flexion power (w/kg)	1.85 (1.44)	5.62 (1.11)	3.27 (1.93)	5.09 (1.41)	0.316	0.077	0.000 *	0.877	0.002 *
Peak hip extension power (w/kg)	5.83 (1.09)	4.72 (0.74)	6.25 (2.00)	3.73 (1.50)	0.443	0.046	0.000 *	0.711	0.114

* Significance (*p* < 0.05); significant interaction between non-dominant/dominant direction and anticipated (A)/unanticipated (UnA). SD—standard deviation; Deg—degrees; N—Newton; m—meter; kg—kilogram; M—moment joint; ω—angular velocity; J—Joule; s—second.

## Data Availability

The data that support the findings of this study are available upon reasonable request from the corresponding author.

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
