# Peer review of "Biomechanic Differences Between Anticipated and Unanticipated Volleyball Block Jump: Implications for Lower Limb Injury Risk"

_life, 2024, doi:10.3390/life14111357_

Round 1

Reviewer 1 Report

Comments and Suggestions for Authors

1)While you mention the high incidence of knee injuries in volleyball, providing specific statistics or citing recent epidemiological studies would strengthen your argument.

2)Provide more details on participant selection criteria. Were there any exclusion criteria beyond being injury-free? How was "injury-free" defined and assessed?

3)Clarify how the unanticipated tasks were operationalized. For example, how much time did participants have to react to the light cues? Were the light cues randomized, and if so, how was randomization achieved?

4)Describe how you controlled for potential confounding variables such as fatigue, learning effects, or motivation levels.

5)Expand on the potential biomechanical explanations for the observed differences between dominant and non-dominant directions and between anticipated and unanticipated tasks.

6)Strengthen the discussion on how your findings relate to specific injury mechanisms, such as anterior cruciate ligament (ACL) injuries or patellar tendinitis.

Comments on the Quality of English Language

The manuscript contains grammatical errors and awkward phrasing that may impede understanding. I recommend thorough proofreading and, if possible, professional language editing to improve readability. Ensure that all abbreviations and acronyms are defined upon first use (e.g., VBJ for Volleyball Block Jump).

Author Response

Response to the reviewers' comments

Dear Editors and Reviewers:

Thank you very much again for your constructive comments, and time spent analyzing our Manuscript entitled "Biomechanic Differences Between Anticipated and Unanticipated Volleyball Block Jump: Implications for Lower Limb In-jury Risk". Those comments are all valuable and very helpful for revising and improving our paper, as well as the important guiding significance to our research. We have studied the comments carefully and made corrections which we hope meet with approval. Revised portions are highlighted in red on the paper. Here below is our description of the revision according to the reviewers' comments.

Reviewer 1

Dear Authors,

  1. While you mention the high incidence of knee injuries in volleyball, providing specific statistics or citing recent epidemiological studies would strengthen your argument.

Response: Many thanks for your constructive comments. We appreciate your feedback and have revised the Introduction based on your suggestions. We conducted a thorough review of the literature and incorporated relevant data to strengthen our argument. For instance, we have conducted a thorough review of the literature and included relevant data to strengthen our argument. For instance, we now cite a recent study indicating that knee injuries account for approximately 30% of all injuries in volleyball players, with anterior cruciate ligament (ACL) injuries being particularly prevalent. This statistical evidence underscores the significance of our research and enhances the context for discussing injury mechanisms. We have revised the Introduction accordingly to ensure it is both comprehensive and compelling.

Revisions: Lines 42-49

Epidemiological studies indicate that approximately 90% of injuries in volleyball occur in the lower limbs [12]. Especially, the incidence of knee injuries, including non-contact ACL ruptures, osteoarthritis, articular cartilage damage, and patellar tendinitis. For instance, epidemiological studies have reported that knee injuries account for up to 30% of all volleyball-related injuries, with non-contact ACL injuries comprising a significant portion of these cases. The recovery process is not only lengthy but also financially burdensome, often requiring extended rehabilitation periods, surgery, and long-term management strategies [13-15].

  1. Provide more details on participant selection criteria. Were there any exclusion criteria beyond being injury-free? How was "injury-free" defined and assessed?

Response: Thank you for your valuable feedback on participant selection criteria. We appreciate your emphasis on clarity. In our study, we established specific eligibility criteria to ensure a homogeneous sample: (1) no significant musculoskeletal injuries; (2) no prior lower limb or spine surgeries; (3) no new injuries within the past year; (4) documentation or self-report of any injuries in the past two years; (5) a minimum of two years of volleyball experience; and (6) no neurological disorders affecting motor skills. This rigorous selection process ensured that all participants were injury-free and capable of performing the required tasks. We hope this clarifies our approach and enhances the quality of our research.

Revisions: Lines 165-172

The eligible criteria for subjects were as follows: (1) no history of significant musculoskeletal injuries, including ligament tears, fractures, tendon injuries, or chronic joint pain; (2) no prior surgical procedures related to the lower limbs or spine; (3) no new musculoskeletal injuries within the past year; (4) for participants with injuries within the past two years, documentation or self-report of injury severity must be provided; (5) a minimum of two years of volleyball playing experience to ensure familiarity with the sport; (6) no neurological disorders that could impact motor skills or coordination [6].

  1. Clarify how the unanticipated tasks were operationalized. For example, how much time did participants have to react to the light cues? Were the light cues randomized, and if so, how was randomization achieved?

Response: Thank you for your valuable feedback regarding the operationalization of unanticipated tasks. To clarify, "unanticipated" means that participants began their blocking approach immediately upon the illumination of one of the three lights, requiring a reaction without prior planning. The light cues were randomized using a computer algorithm to maintain unpredictability. In both anticipated and unanticipated conditions, participants aimed to reach the designated location as quickly as possible, with the anticipated condition simulating typical training scenarios. We hope this additional detail addresses your concerns.

Revisions: Lines 136-142

The term 'unanticipated' refers to the immediate initiation of the blocking approach upon the illumination of one of the three lights, requiring participants to react without the opportunity for conscious planning. The light cues were randomized using a computer algorithm to ensure unpredictability in their sequence. In both the anticipated and unanticipated conditions, participants were instructed to reach the designated location as quickly as possible. The anticipated condition mirrored typical training scenarios, such as daily ball-free blocking drills or situations with a single attacker.

  1. Describe how you controlled for potential confounding variables such as fatigue, learning effects, or motivation levels.

Response: Many thanks for your comments. In the revised manuscript, we have added a detailed description of how we controlled factors such as fatigue, learning effects, and motivation levels. We specified that participants underwent a familiarization session to minimize learning effects. We have added detailed information regarding these controls to the revised manuscript to enhance clarity. Thank you again for your helpful suggestions.

Revisions: Lines 129-134

To ensure the integrity of the results, we implemented several strategies to control potential confounding variables. Fatigue was minimized through adequate rest periods between trials and a standardized warm-up routine, while practice trials were incorporated to mitigate learning effects, allowing participants to familiarize themselves with the tasks and equipment. These strategies created a more reliable testing environment.

  1. Expand on the potential biomechanical explanations for the observed differences between dominant and non-dominant directions and between anticipated and unanticipated tasks.

Response: Many thanks for your comments. We have enhanced our discussion to clarify the biomechanical differences in movement patterns during the block jump. Specifically, we explain that the non-dominant direction's increased hip and knee flexion plays a vital role in absorbing impact forces, thereby reducing injury risk. Conversely, the dominant direction's more rigid posture is critical for maintaining forward momentum. Your input has significantly contributed to this refinement.

Revisions: Lines 317-325

These biomechanical differences in movement patterns can be attributed to the specific demands placed on each limb during the block jump. In the non-dominant direction, the increased flexion at the hip and knee is crucial for absorbing impact forces, effectively mitigating injury risk. In contrast, the dominant direction maintains a more rigid posture, which is essential for sustaining forward momentum. Importantly, our findings align with prior studies that the non-dominant direction results in a heightened peak valgus moment at the knee. This is particularly significant, as numerous studies have established that larger peak valgus moments are associated with an increased risk of knee injuries [43].

  1. Strengthen the discussion on how your findings relate to specific injury mechanisms, such as anterior cruciate ligament (ACL) injuries or patellar tendinitis.

Response: Thank you for your valuable feedback. We have refined our discussion to better connect our findings with specific injury mechanisms. We highlighted how rapid extension during jumping can lead to forceful landings, increasing mechanical stress on the knee joint. This elevated peak ground reaction force (pGRF) in the dominant direction may contribute to ACL injury risk through excessive knee valgus and shear forces. We also mentioned the potential for overuse injuries like patellar tendinitis. We appreciate your guidance in strengthening this aspect of our study.

Revisions: Lines 335-343

This rapid extension can lead to more forceful landings, imposing significant mechanical stress on the knee joint. The elevated pGRF observed in the dominant direction indicates that the dominant limb bears greater loads upon landing. Such heightened forces are critical in the mechanisms underlying anterior cruciate ligament (ACL) injuries, as they may induce excessive knee valgus and anterior shear forces on the tibia, thereby increasing the risk of ligament strain or rupture [46,47]. Furthermore, the dynamic nature of these movements is associated with an elevated risk of overuse injuries, such as patellar tendinitis, particularly when the knee is subjected to repetitive high forces without adequate recovery [48].

  1. Comments on the Quality of English Language:

The manuscript contains grammatical errors and awkward phrasing that may impede understanding. I recommend thorough proofreading and, if possible, professional language editing to improve readability. Ensure that all abbreviations and acronyms are defined upon first use (e.g., VBJ for Volleyball Block Jump).

Response: Thank you for your constructive comments regarding the quality of the English language in our manuscript. We appreciate your attention to detail and will undertake a comprehensive proofreading process to address grammatical errors and awkward phrasing. Additionally, we will ensure that all abbreviations and acronyms are clearly defined upon first use. We have already sought the assistance of a native English speaker for proofreading and editing, which has significantly improved the clarity and readability of the manuscript. Thank you very much!

Thank you again for your constructive and valuable criticism, and the time spent analyzing this Manuscript. I reiterate our sincere gratitude to you for your help and patience. Thank you very much!

Reviewer 2 Report

Comments and Suggestions for Authors

Dear Authors,

The data and their form of presentation is optimal. I congratulate the authors for an interesting article, which brings useful information to those in the field of performance sports, but also to those who work in rehabilitation, in the recovery of locomotor disorders in athletes.

Below are some aspects of the acre I would like additional information on:

- I didn't understand why in figure no. 1 there are photos of women and the subjects are men?!! Did the authors not specify that the subjects were male? Wouldn't he have found it, from so many recordings and images of men???

- the criteria for inclusion, but especially those for excluding subjects, should be better detailed. In their current form, they are a bit vague

-L150 "All the subjects were right-handed"- I think this aspect should be clarified very, very well. What do the authors want to say through this aspect?, is it not clear?!!!! because "right-handed" hitting does not necessarily imply left cerebral hemispheric dominance. Also, "right-handed" as dominant does not justify dominance for the lower limbs. in no way. I ask the authors to elaborate or clearly argue this point. The dominance of the right hand is associated with the dominance of the left lower limb, and the specification "All the subjects were right-handed" did not clarify me, on the contrary, it confused me, because the authors do not clarify here the relationship between "right-handed" and "left lower limb".

- table 1 and 3 should be rearranged, in their current form they do not provide a clear understanding, because the data are piled up and do not provide an overview of the differences in the evaluated indicators;

Otherwise, an ok, interesting and useful article, especially for volleyball enthusiasts.

Success

Author Response

Response to the reviewers' comments

Dear Editors and Reviewers:

Thank you very much again for your constructive comments, and time spent analyzing our Manuscript entitled "Biomechanic Differences Between Anticipated and Unanticipated Volleyball Block Jump: Implications for Lower Limb In-jury Risk". Those comments are all valuable and very helpful for revising and improving our paper, as well as the important guiding significance to our research. We have studied the comments carefully and made corrections which we hope meet with approval. Revised portions are highlighted in red on the paper. Here below is our description of the revision according to the reviewers' comments.

Reviewer 2

Dear Authors,

The data and their form of presentation is optimal. I congratulate the authors for an interesting article, which brings useful information to those in the field of performance sports, but also to those who work in rehabilitation, in the recovery of locomotor disorders in athletes.

Response: Thank you very much for your kind words and positive feedback on our article. We are thrilled to hear that you found the data presentation optimal and that the information provided is valuable to both performance sports and rehabilitation professionals. Your encouragement is greatly appreciated, and it motivates us to continue our research in this important area. Thank you again!

  1. I didn't understand why in figure no. 1 there are photos of women and the subjects are men?!! Did the authors not specify that the subjects were male? Wouldn't he have found it, from so many recordings and images of men???

Response: Thank you for your insightful comment. We sincerely apologize for the oversight regarding the images in Figure 1. We have corrected this issue by updating the figure to ensure it aligns with the specifications of our study. We appreciate your understanding and attention to detail, and we thank you for bringing this to our attention.

Revisions: Lines 152-157

Figure 1. The red box is the attacker, the green box is the second passer, and the yellow box is the blocker. The anticipated situation can be interpreted as the blocker can determine where the attacker is spiking the ball from, as in the left picture where there is only one attacker. In the unanticipated case, the blocker cannot tell who is going to attack, as, in the right picture, three blockers must face four attackers.

  1. the criteria for inclusion, but especially those for excluding subjects, should be better detailed. In their current form, they are a bit vague

Response: Thank you for your valuable feedback regarding participant selection criteria. We appreciate your emphasis on clarity and comprehensiveness. In our study, we established specific eligibility criteria to ensure a homogeneous and representative sample. Participants were required to meet the following conditions: (1) no history of significant musculoskeletal injuries, such as ligament tears, fractures, tendon injuries, or chronic joint pain; (2) no prior surgical procedures related to the lower limbs or spine; (3) no new musculoskeletal injuries within the past year; (4) for those with injuries in the past two years, we required documentation or self-reporting of injury severity; (5) a minimum of two years of volleyball playing experience to ensure familiarity with the sport; and (6) no neurological disorders that could impact motor skills or coordination. This rigorous selection process was designed to ensure that all participants were injury-free and capable of performing the tasks required in our study. We hope this clarifies our approach and strengthens the overall quality of our research.

Revisions: Lines 165-172

The eligible criteria for subjects were as follows: (1) no history of significant musculoskeletal injuries, including ligament tears, fractures, tendon injuries, or chronic joint pain; (2) no prior surgical procedures related to the lower limbs or spine; (3) no new musculoskeletal injuries within the past year; (4) for participants with injuries within the past two years, documentation or self-report of injury severity must be provided; (5) a minimum of two years of volleyball playing experience to ensure familiarity with the sport; (6) no neurological disorders that could impact motor skills or coordination [6].

  1. L150 "All the subjects were right-handed"- I think this aspect should be clarified very, very well. What do the authors want to say through this aspect?, is it not clear?!!!! because "right-handed" hitting does not necessarily imply left cerebral hemispheric dominance. Also, "right-handed" as dominant does not justify dominance for the lower limbs. in no way. I ask the authors to elaborate or clearly argue this point. The dominance of the right hand is associated with the dominance of the left lower limb, and the specification "All the subjects were right-handed" did not clarify me, on the contrary, it confused me, because the authors do not clarify here the relationship between "right-handed" and "left lower limb".

Response: Thank you for your insightful comment regarding the clarification of limb dominance. We appreciate your attention to this detail. To clarify, the dominant limb in our study was defined as the preferred leg for tasks such as kicking a ball. All participants exhibited right-leg dominance, identified as their preferred leg for power generation during take-off and blocking movements. This distinction is important, as the dominant leg typically experiences greater loads during explosive actions and is crucial for stabilizing the body in critical phases of play. We will revise the manuscript to ensure this explanation is clear and comprehensive. Your feedback is invaluable in enhancing the clarity of our work!

Revisions: Lines 122-127

The dominant limb was defined as the preferred leg for tasks such as kicking a ball. In this study, all participants exhibited right-leg dominance, identified as their preferred leg for power generation during take-off and directional movements in blocking. This assessment of lower limb dominance is essential for understanding movement efficiency and injury risk, as the dominant leg bears a greater load during explosive actions and stabilizes the body in critical phases of play [6,36].

  1. table 1 and 3 should be rearranged, in their current form they do not provide a clear understanding, because the data are piled up and do not provide an overview of the differences in the evaluated indicators;

Response: Many thanks for your comments. We apologize for any confusion caused by their current layout. We will rearrange the tables to improve clarity and facilitate a better understanding of the differences in the evaluated indicators. Your suggestions are invaluable, and we appreciate your attention to detail as we enhance the presentation of our data.

Revisions: Lines 257-261; Lines 295-299

Table 1. Kinematic variables for the dominant limb joints during block jumping (mean ± standard deviation)

Anticipated

Unanticipated

Anova p-value

DD

(SD)

NDD

(SD)

DD

(SD)

NDD

(SD)

p-value A v UnA

Effect Size

p-value D

Effect Size

Interaction

Jump height (m)

0.58

(0.09)

0.57

(0.10)

0.57

(0.09)

0.60

(0.08)

0.444

0.020

0.618

0.009

0.240

Approaching velocity (m/s)

0.38

(0.05)

0.39

(0.04)

0.41

(0.12)

0.40

(0.09)

0.562

0.011

0.715

0.007

0.362

Ankle

Ankle angle at contact (deg)

8.17

(5.42)

7.33

(6.85)

6.17

(6.72)

6.12

(6.57)

0.110

0.099

0.696

0.006

0.803

Peak ankle plantar-flexion angle (deg)

26.83

(5.62)

26.24

(6.77)

27.89

(7.26)

29.27

(7.04)

0.413

0.052

0.780

0.006

0.450

Peak ankle dorsiflexion angle (deg)

26.37

(6.22)

34.61

(4.65)

29.80

(2.76)

32.54

(5.28)

0.643

0.017

0.001*

0.706

0.081

Peak ankle dorsiflexion velocity (deg/s)

121.37

(36.71)

344.13

(71.01)

109.64

(44.95)

251.63

(60.12)

0.001*

0.577

0.000*

0.948

0.055

Peak ankle plantar-flexion velocity (deg/s)

468.62

(40.45)

171.54

(29.66)

390.75

(83.31)

164.34

(29.85)

0.014*

0.385

0.000*

0.978

0.026*

Knee

Knee angle at contact (deg)

64.71

(16.17)

40.33

(9.45)

70.25

(8.06)

42.62

(12.11)

0.454

0.044

0.000*

0.929

0.417

Peak knee flexion angle (deg)

81.60

(16.00)

86.72

(4.36)

79.66

(13.00)

94.79

(13.61)

0.517

0.033

0.004*

0.490

0.126

Peak knee flexion velocity (deg/s)

89.86

(46.98)

356.92

(107.84)

78.12

(57.02)

367.04

(116.05)

0.965

0.000

0.000*

0.892

0.611

Peak knee extension velocity (deg/s)

437.83

(104.76)

306.17

(45.59)

309.64

(156.81)

284.11

(31.70)

0.063

0.242

0.000*

0.684

0.086

Peak knee abduction angle (deg)

2.56

(4.39)

1.34

(3.42)

2.54

(4.95)

2.54

(4.95)

0.769

0.007

0.330

0.079

0.452

Hip

Peak hip flexion angle (deg)

57.59

(15.94)

67.26

(18.22)

58.53

(13.56)

66.90

(16.14)

0.959

0.000

0.012*

0.394

0.801

Peak hip flexion velocity (deg/s)

213.76

(54.49)

85.18

(14.39)

185.74

(40.83)

72.50

(22.70)

0.189

0.139

0.000*

0.960

0.491

* Significance (p<0.05); Significant interaction between Non-Dominant /Dominant direction and Anticipated (A)/Unanticipated (UnA).SD-standard deviation; Deg–degrees; N–Newton; m- meter; kg–kilogram; M–Moment Joint; ω–angular velocity; J–Joule; s–second.

Table 3. Kinetic variables for the dominant limb joints during a block jumping (mean ± standard deviation).

Anticipated

Unanticipated

Anova p-value

DD

(SD)

NDD

(SD)

DD

(SD)

NDD

(SD)

p-value A v UnA

Effect Size

p-value D

Effect Size

Interaction

Ankle

Peak ankle plantar-flexion moment (Nm/kg)

0.02

(0.05)

0.04

(0.06)

0.02

(0.02)

0.04

(0.05)

0.725

0.011

0.251

0.108

0.982

Peak ankle dorsiflexion moment (Nm/kg)

-2.25

(0.41)

2.12

(0.32)

2.33

(0.50)

2.12

(0.31)

0.725

0.011

0.069

0.250

0.662

Peak ankle plantar-flexion power (w/kg)

16.10

(2.75)

16.71

(3.34)

16.99

(3.65)

17.49

(3.09)

0.528

0.031

0.405

0.054

0.933

Peak ankle dorsiflexion power (w/kg)

3.13

(1.24)

3.61

(0.73)

4.02

(1.71)

3.34

(0.72)

0.389

0.058

0.719

0.010

0.097

Knee

Peak knee flexion moment (Nm/kg)

0.31

(0.19)

0.50

(0.31)

0.25

(0.18)

0.39

(0.20)

0.421

0.055

0.002*

0.548

0.456

Peak knee extension moment (Nm/kg)

2.85

(0.22)

2.94

(0.32)

2.81

(0.26)

2.96

(0.29)

0.908

0.001

0.120

0.189

0.632

Peak knee abduction moment (Nm/kg)

0.12

(0.18)

0.54

(0.27)

0.15

(0.10)

0.40

(0.38)

0.077

0.238

0.001*

0.617

0.629

Peak knee flexion power(w/kg)

6.68

(1.49)

9.88

(2.09)

5.47

(2.39)

9.08

(1.90)

0.027*

0.346

0.000*

0.666

0.734

Peak knee extension power(w/kg)

18.35

(1.92)

15.94

(1.35)

19.21

(2.24)

17.32

(2.21)

0.081

0.232

0.005*

0.499

0.671

Proximal anterior tibia shear force (BW)

0.55

(0.31)

0.47

(0.28)

0.43

(0.26)

0.41

(0.18)

0.194

0.117

0.292

0.079

0.606

Hip

Peak hip flexion moment (Nm/kg)

2.61

(0.64)

2.88

(1.06)

2.74

(1.04)

3.00

(0.86)

0.565

0.028

0.202

0.132

0.992

Peak hip extension moment (Nm/kg)

0.46

(0.23)

0.62

(0.25)

0.71

(0.34)

0.59

(0.18)

0.083

0.214

0.847

0.003

0.027*

Peak hip flexion power (w/kg)

1.85

(1.44)

5.62

(1.11)

3.27

(1.93)

5.09

(1.41)

0.316

0.077

0.000*

0.877

0.002*

Peak hip extension power (w/kg)

5.83

(1.09)

4.72

(0.74)

6.25

(2.00)

3.73

(1.50)

0.443

0.046

0.000*

0.711

0.114

* Significance (p<0.05); Significant interaction between Non-Dominant /Dominant direction and Anticipated (A)/Unanticipated (UnA). SD-standard deviation; Deg–degrees; N–Newton; m- meter; kg–kilogram; M–Moment Joint; ω–angular velocity; J–Joule; s–second.

  1. Otherwise, an ok, interesting and useful article, especially for volleyball enthusiasts. Success

Response: Thank you very much for your positive feedback! We're glad to hear that you found the article interesting and useful, especially for volleyball enthusiasts. We appreciate your support and encouragement as we continue to refine our work.

Thank you again for your constructive and valuable criticism, and the time spent analyzing this Manuscript. I reiterate our sincere gratitude to you for your help and patience. Thank you very much!

Reviewer 3 Report

Comments and Suggestions for Authors

I consider the presented scientific work to be rigorous and to include all the necessary investigative elements pertinent to the field. It describes the current level of research in the field of sports, highlighting the lack of scientific investigation into the causes of injuries among volleyball players. The study focuses on a group of athletes (testing cohort) performing gameplay scenarios that may lead to injuries. Specifically, the study centers on comparing dominant versus non-dominant movements, as well as anticipated versus unanticipated actions. The results obtained through the use of industry-standard data acquisition equipment, which records the kinematics and dynamics of movements, and muscle loading, and generates simulations of athletes' movements, are presented. In conclusion, the paper identifies the overstrained muscle regions associated with these movements, which may lead to injury. Additionally, one of the neural theories that may explain this type of action is mentioned.

I suggest that the authors be more specific when presenting neural theories (e.g., top-down control - general motivation). Common causes of injury include excessive muscular loading, sudden movements without prior warm-up, muscle fatigue, muscle weakness or structural deficiencies, direct trauma, lack of flexibility and elasticity, repetitive loading and overuse, inadequate recovery, individual factors, and age. Most of these factors have been excluded through the selection of the test group and investigative methods. In the presented case, the determining factor remains excessive muscular loading. In unanticipated situations, athletes quickly resort to automatic systems to execute the necessary movements, and attention shifts entirely to achieving the primary goal. This rapid transition can lead to overloading the muscular system and ignoring safety-related sensory feedback, increasing the risk of injury. The actions described in the paper (which push the athlete to and beyond physical limits, leading to injury) can be explained by several neural theories. A few examples include Attention Resource Theory, Goal Prioritization Theory, Central Resource Theory, and especially Dual-Process Theory, or Goal Prioritization and Risk Perception Theory.

I also recommend that the Conclusions section be more specific in providing practical guidance for athletes and coaches to avoid such situations.

It would be advisable for the paper to go beyond merely presenting the results of data acquisition.

Author Response

Response to the reviewers' comments

Dear Editors and Reviewers:

Thank you very much again for your constructive comments, and time spent analyzing our Manuscript entitled "Biomechanic Differences Between Anticipated and Unanticipated Volleyball Block Jump: Implications for Lower Limb In-jury Risk". Those comments are all valuable and very helpful for revising and improving our paper, as well as the important guiding significance to our research. We have studied the comments carefully and made corrections which we hope meet with approval. Revised portions are highlighted in red on the paper. Here below is our description of the revision according to the reviewers' comments.

Reviewer 3

Dear authors,

I consider the presented scientific work to be rigorous and to include all the necessary investigative elements pertinent to the field. It describes the current level of research in the field of sports, highlighting the lack of scientific investigation into the causes of injuries among volleyball players. The study focuses on a group of athletes (testing cohort) performing gameplay scenarios that may lead to injuries. Specifically, the study centers on comparing dominant versus non-dominant movements, as well as anticipated versus unanticipated actions. The results obtained through the use of industry-standard data acquisition equipment, which records the kinematics and dynamics of movements, and muscle loading, and generates simulations of athletes' movements, are presented. In conclusion, the paper identifies the overstrained muscle regions associated with these movements, which may lead to injury. Additionally, one of the neural theories that may explain this type of action is mentioned.

Response: Thank you for your encouraging feedback on our study. We appreciate your recognition of the rigor and focus on injury mechanisms in volleyball. Your acknowledgment of our data collection methods and insights into overstrained muscle regions is greatly appreciated. We also value your mention of the neural theory as a potential explanation. We look forward to further refining our work based on your suggestions.

  1. I suggest that the authors be more specific when presenting neural theories (e.g., top-down control - general motivation). Common causes of injury include excessive muscular loading, sudden movements without prior warm-up, muscle fatigue, muscle weakness or structural deficiencies, direct trauma, lack of flexibility and elasticity, repetitive loading and overuse, inadequate recovery, individual factors, and age. Most of these factors have been excluded through the selection of the test group and investigative methods. In the presented case, the determining factor remains excessive muscular loading. In unanticipated situations, athletes quickly resort to automatic systems to execute the necessary movements, and attention shifts entirely to achieving the primary goal. This rapid transition can lead to overloading the muscular system and ignoring safety-related sensory feedback, increasing the risk of injury. The actions described in the paper (which push the athlete to and beyond physical limits, leading to injury) can be explained by several neural theories. A few examples include Attention Resource Theory, Goal Prioritization Theory, Central Resource Theory, and especially Dual-Process Theory, or Goal Prioritization and Risk Perception Theory.

Response: Thank you for your valuable suggestion regarding the presentation of neural theories. We have revised the manuscript to incorporate a more specific discussion of neural control mechanisms. In particular, we have highlighted how top-down control processes operate under anticipated conditions, enabling conscious movement regulation. In contrast, unanticipated scenarios invoke automatic neuromuscular processes that prioritize rapid goal achievement, often at the expense of sensory feedback. This transition is explained by the Attention Resource Theory, where the rapid reallocation of neural resources in unanticipated situations can lead to excessive muscular loading, thus increasing injury risk. We have included references to these neural theories to provide a clearer and more precise theoretical framework for understanding the mechanisms behind injury risk in these contexts. We appreciate your insight and believe these additions improve the clarity and depth of our discussion.

Revisions: Lines 346-354

This shift can be understood through neural control theories, which describe how under anticipated conditions, top-down control mechanisms are employed, allowing athletes to consciously regulate movements. However, in unanticipated scenarios, the neuromuscular system switches to automatic processes that prioritize immediate goal achievement over sensory feedback, as explained by the Attention Resource Theory. The rapid reallocation of neural resources under these conditions can lead to excessive muscular loading, increasing the risk of injury due to a diminished capacity to process proprioceptive signals that would normally adjust movements to prevent overload [51].

  1. I also recommend that the Conclusions section be more specific in providing practical guidance for athletes and coaches to avoid such situations.

It would be advisable for the paper to go beyond merely presenting the results of data acquisition.

Response: Thank you for your constructive feedback on the Conclusions section. We agree that providing practical guidance is essential. In response, we have revised the Conclusions to offer more specific recommendations for athletes and coaches. Specifically, we emphasize the importance of incorporating both anticipated and unanticipated scenarios into training regimens to improve athletes’ adaptability. Additionally, we suggest targeted exercises to strengthen key muscle groups and enhance neuromuscular control, which could help mitigate the risk of injury due to excessive muscular loading in unanticipated situations. These practical recommendations are designed to bridge the gap between data acquisition results and real-world applications, providing actionable insights to reduce injury risks during high-intensity sports like volleyball. We believe these revisions will make the study more applicable to both sports practitioners and researchers. Thank you once again for your helpful suggestions.

Revisions: Lines 392-398

In practical applications, coaches should consider the differences in limb coordination and movement directions by incorporating unilateral preventive exercises. Moreover, simulating competitive scenarios during training can better prepare athletes for unanticipated situations, fostering quicker decision-making and adaptive movement strategies. By enhancing athletes' responsiveness and movement efficiency in unpredictable contexts, these training methods may significantly reduce the risk of injuries associated with excessive muscular loading.

Thank you again for your constructive and valuable criticism, and the time spent analyzing this Manuscript. I reiterate our sincere gratitude to you for your help and patience. Thank you very much!

Reviewer 4 Report

Comments and Suggestions for Authors

Dear Authors

Introduction The paper provides a valuable contribution to volleyball injury prevention by analyzing lower limb biomechanics in block jumping, comparing dominant and non-dominant directions under both anticipated and unanticipated conditions.    In the abstract and throughout the paper there a many minor grammatical errors. Please read through again and make corrections.     Method The study’s thorough investigation into joint kinematics and kinetics, such as ankle dorsiflexion and knee moments, offers critical insights for understanding injury risk, particularly the stiffer movement patterns in the dominant direction and increased knee load in the non-dominant.   Results The findings are practically relevant, with recommendations for targeted training programs. However, the small sample size, lack of muscle activation data, and limited discussion of broader applications and training specifics constrain the study's generalizability and impact.   A number of the columns are very squashed and I suggest orientating the page to landscape where required.    For Figure 4, provide an elaboration of the acronym UNA.   Discussion Expanding the scope to include a more diverse athlete population and offering deeper injury mechanism analysis would enhance its utility in sports settings.   Include a strength and limitations section in the discussion.    Kindest regards

Comments on the Quality of English Language

The paper contains minor grammatical errors, including inconsistent use of tense, lack of commas in complex sentences, and occasional awkward phrasing, which slightly detract from the overall readability but do not significantly affect the clarity of the research findings

Author Response

Response to the reviewers' comments

Dear Editors and Reviewers:

Thank you very much again for your constructive comments, and time spent analyzing our Manuscript entitled "Biomechanic Differences Between Anticipated and Unanticipated Volleyball Block Jump: Implications for Lower Limb In-jury Risk". Those comments are all valuable and very helpful for revising and improving our paper, as well as the important guiding significance to our research. We have studied the comments carefully and made corrections which we hope meet with approval. Revised portions are highlighted in red on the paper. Here below is our description of the revision according to the reviewers' comments.

Reviewer 4

Dear authors,

  1. Introduction

The paper provides a valuable contribution to volleyball injury prevention by analyzing lower limb biomechanics in block jumping, comparing dominant and non-dominant directions under both anticipated and unanticipated conditions. In the abstract and throughout the paper there a many minor grammatical errors. Please read through again and make corrections.

Response: Thank you for your thoughtful feedback on the contribution of our paper to volleyball injury prevention. We appreciate your recognition of our focus on lower limb biomechanics in block jumping. We acknowledge the presence of grammatical errors and will conduct a thorough review of the manuscript to correct these issues. Your attention to detail is greatly appreciated, and we are committed to enhancing the clarity and quality of our work. Thank you once again for your valuable input!

Revisions: Lines 11-31

Abstract: Volleyball is a high-intensity sport characterized by repetitive jumping, sudden directional changes, and overhead movements, all of which significantly increase the risk of injuries, particularly to the shoulders, knees, and ankles. Despite the frequency of injuries caused by actions like blocking and spiking, there has been limited research focused on the specific biomechanical risk factors unique to volleyball. This study aimed to investigate the lower limb biomechanics during block jumps in both the dominant and non-dominant directions, under both anticipated and unanticipated conditions, in fifteen elite male volleyball players. Kinematic and kinetic data from the ankle, knee, and hip joints were recorded. Our results revealed statistically significant differences between the dominant and non-dominant directions at the ankle, knee, and hip joints. The non-dominant direction exhibited a greater ankle dorsiflexion angle and velocity, as well as a higher knee flexion angle, velocity, moment, power, and abduction moment, along with increased hip flexion angle and power. Additionally, unanticipated movements led to increases in vertical ground reaction force (vGRF), hip extension moment, and flexion power, while ankle dorsiflexion plantarflexion velocity, and knee flexion power decreased. It appears that movements in the dominant direction were stiffer and less cushioned, potentially increasing the risk of injury. While the non-dominant direction provided better shock absorption, it also elevated the knee valgus moment, which could increase the load on the knee. Furthermore, in unanticipated situations, athletes with short reaction times, unable to quickly adjust their automated movement patterns, faced a higher risk of limb overuse, thereby increasing the likelihood of injury. In practice, coaches should consider differences in limb coordination and movement direction, incorporating unilateral preventive exercises to reduce the risk of injury.

  1. Method

The study’s thorough investigation into joint kinematics and kinetics, such as ankle dorsiflexion and knee moments, offers critical insights for understanding injury risk, particularly the stiffer movement patterns in the dominant direction and increased knee load in the non-dominant.

Response: Thank you for your positive remarks regarding our study's investigation into joint kinematics and kinetics. We appreciate your acknowledgment of the insights provided, particularly regarding the stiffer movement patterns in the dominant direction and the increased knee load in the non-dominant direction. These findings are indeed crucial for understanding injury risk in athletes, and we hope they contribute to improved prevention strategies in sports. Your feedback is greatly appreciated!

  1. Results

The findings are practically relevant, with recommendations for targeted training programs. However, the small sample size, lack of muscle activation data, and limited discussion of broader applications and training specifics constrain the study's generalizability and impact. A number of the columns are very squashed and I suggest orientating the page to landscape where required. For Figure 4, provide an elaboration of the acronym UNA.

Response:

Thank you for your valuable feedback. We sincerely apologize for the lack of muscle activation data in our study. We have acknowledged this limitation in the revised limitations section and hope that future research can address this gap to provide a more comprehensive understanding of the biomechanics involved. Additionally, we will address the formatting issues in the tables and provide a clearer explanation of the acronym "UNA" in Figure 4. Your insights are invaluable, and we thank you for bringing these matters to our attention!

Revisions: Lines 257-261; Lines 295-299; Lines 363-374

Table 1. Kinematic variables for the dominant limb joints during block jumping (mean ± standard deviation)

Anticipated

Unanticipated

Anova p-value

DD

(SD)

NDD

(SD)

DD

(SD)

NDD

(SD)

p-value A v UnA

Effect Size

p-value D

Effect Size

Interaction

Jump height (m)

0.58

(0.09)

0.57

(0.10)

0.57

(0.09)

0.60

(0.08)

0.444

0.020

0.618

0.009

0.240

Approaching velocity (m/s)

0.38

(0.05)

0.39

(0.04)

0.41

(0.12)

0.40

(0.09)

0.562

0.011

0.715

0.007

0.362

Ankle

Ankle angle at contact (deg)

8.17

(5.42)

7.33

(6.85)

6.17

(6.72)

6.12

(6.57)

0.110

0.099

0.696

0.006

0.803

Peak ankle plantar-flexion angle (deg)

26.83

(5.62)

26.24

(6.77)

27.89

(7.26)

29.27

(7.04)

0.413

0.052

0.780

0.006

0.450

Peak ankle dorsiflexion angle (deg)

26.37

(6.22)

34.61

(4.65)

29.80

(2.76)

32.54

(5.28)

0.643

0.017

0.001*

0.706

0.081

Peak ankle dorsiflexion velocity (deg/s)

121.37

(36.71)

344.13

(71.01)

109.64

(44.95)

251.63

(60.12)

0.001*

0.577

0.000*

0.948

0.055

Peak ankle plantar-flexion velocity (deg/s)

468.62

(40.45)

171.54

(29.66)

390.75

(83.31)

164.34

(29.85)

0.014*

0.385

0.000*

0.978

0.026*

Knee

Knee angle at contact (deg)

64.71

(16.17)

40.33

(9.45)

70.25

(8.06)

42.62

(12.11)

0.454

0.044

0.000*

0.929

0.417

Peak knee flexion angle (deg)

81.60

(16.00)

86.72

(4.36)

79.66

(13.00)

94.79

(13.61)

0.517

0.033

0.004*

0.490

0.126

Peak knee flexion velocity (deg/s)

89.86

(46.98)

356.92

(107.84)

78.12

(57.02)

367.04

(116.05)

0.965

0.000

0.000*

0.892

0.611

Peak knee extension velocity (deg/s)

437.83

(104.76)

306.17

(45.59)

309.64

(156.81)

284.11

(31.70)

0.063

0.242

0.000*

0.684

0.086

Peak knee abduction angle (deg)

2.56

(4.39)

1.34

(3.42)

2.54

(4.95)

2.54

(4.95)

0.769

0.007

0.330

0.079

0.452

Hip

Peak hip flexion angle (deg)

57.59

(15.94)

67.26

(18.22)

58.53

(13.56)

66.90

(16.14)

0.959

0.000

0.012*

0.394

0.801

Peak hip flexion velocity (deg/s)

213.76

(54.49)

85.18

(14.39)

185.74

(40.83)

72.50

(22.70)

0.189

0.139

0.000*

0.960

0.491

* Significance (p<0.05); Significant interaction between Non-Dominant /Dominant direction and Anticipated (A)/Unanticipated (UnA).SD-standard deviation; Deg–degrees; N–Newton; m- meter; kg–kilogram; M–Moment Joint; ω–angular velocity; J–Joule; s–second.

Table 3. Kinetic variables for the dominant limb joints during a block jumping (mean ± standard deviation).

Anticipated

Unanticipated

Anova p-value

DD

(SD)

NDD

(SD)

DD

(SD)

NDD

(SD)

p-value A v UnA

Effect Size

p-value D

Effect Size

Interaction

Ankle

Peak ankle plantar-flexion moment (Nm/kg)

0.02

(0.05)

0.04

(0.06)

0.02

(0.02)

0.04

(0.05)

0.725

0.011

0.251

0.108

0.982

Peak ankle dorsiflexion moment (Nm/kg)

-2.25

(0.41)

2.12

(0.32)

2.33

(0.50)

2.12

(0.31)

0.725

0.011

0.069

0.250

0.662

Peak ankle plantar-flexion power (w/kg)

16.10

(2.75)

16.71

(3.34)

16.99

(3.65)

17.49

(3.09)

0.528

0.031

0.405

0.054

0.933

Peak ankle dorsiflexion power (w/kg)

3.13

(1.24)

3.61

(0.73)

4.02

(1.71)

3.34

(0.72)

0.389

0.058

0.719

0.010

0.097

Knee

Peak knee flexion moment (Nm/kg)

0.31

(0.19)

0.50

(0.31)

0.25

(0.18)

0.39

(0.20)

0.421

0.055

0.002*

0.548

0.456

Peak knee extension moment (Nm/kg)

2.85

(0.22)

2.94

(0.32)

2.81

(0.26)

2.96

(0.29)

0.908

0.001

0.120

0.189

0.632

Peak knee abduction moment (Nm/kg)

0.12

(0.18)

0.54

(0.27)

0.15

(0.10)

0.40

(0.38)

0.077

0.238

0.001*

0.617

0.629

Peak knee flexion power(w/kg)

6.68

(1.49)

9.88

(2.09)

5.47

(2.39)

9.08

(1.90)

0.027*

0.346

0.000*

0.666

0.734

Peak knee extension power(w/kg)

18.35

(1.92)

15.94

(1.35)

19.21

(2.24)

17.32

(2.21)

0.081

0.232

0.005*

0.499

0.671

Proximal anterior tibia shear force (BW)

0.55

(0.31)

0.47

(0.28)

0.43

(0.26)

0.41

(0.18)

0.194

0.117

0.292

0.079

0.606

Hip

Peak hip flexion moment (Nm/kg)

2.61

(0.64)

2.88

(1.06)

2.74

(1.04)

3.00

(0.86)

0.565

0.028

0.202

0.132

0.992

Peak hip extension moment (Nm/kg)

0.46

(0.23)

0.62

(0.25)

0.71

(0.34)

0.59

(0.18)

0.083

0.214

0.847

0.003

0.027*

Peak hip flexion power (w/kg)

1.85

(1.44)

5.62

(1.11)

3.27

(1.93)

5.09

(1.41)

0.316

0.077

0.000*

0.877

0.002*

Peak hip extension power (w/kg)

5.83

(1.09)

4.72

(0.74)

6.25

(2.00)

3.73

(1.50)

0.443

0.046

0.000*

0.711

0.114

* Significance (p<0.05); Significant interaction between Non-Dominant /Dominant direction and Anticipated (A)/Unanticipated (UnA). SD-standard deviation; Deg–degrees; N–Newton; m- meter; kg–kilogram; M–Moment Joint; ω–angular velocity; J–Joule; s–second.

Despite its contributions, this study has several limitations. The relatively small sample size, focus on male athletes with uniform blocking techniques, and consideration of only lower limb movements may limit the generalizability of the findings [58]. Additionally, although participants were instructed to move as fast as possible and land with the dominant limb on the force platform, this controlled setup does not fully mirror the complexity of real-game scenarios. A more significant limitation is the absence of muscle activation data, which could provide a deeper understanding of the neuromuscular strategies involved in volleyball block jumping. Future research should aim to simulate more realistic game conditions, incorporating varied stimuli and muscle activation analysis to better understand motor strategies in block jumping [28,58].

  1. Discussion

Expanding the scope to include a more diverse athlete population and offering deeper injury mechanism analysis would enhance its utility in sports settings. Include a strength and limitations section in the discussion.

Response: Many thanks for your comments. We appreciate your suggestion to expand the scope of our study to include a more diverse athlete population, as well as to provide a deeper analysis of injury mechanisms. In response, we have added a strengths and limitations section in the discussion to highlight the key contributions of our research while acknowledging its constraints. This addition aims to enhance the overall utility of the study in practical sports settings. Thank you again for your helpful suggestions.

Revisions: Lines 369-382

In this study, we have created an experimental protocol that simulates a real race environment and integrated lower limb variables in the sagittal plane that have been previously reported as risk factors in lower limb injuries. Additionally, we strictly standardized the approach speed and distance to enhance the ecological validity of the variable differences observed. Despite its contributions, this study has several limitations. The relatively small sample size, focus on male athletes with uniform blocking techniques, and consideration of only lower limb movements may limit the generalizability of the findings [57]. Additionally, although participants were instructed to move as fast as possible and land with the dominant limb on the force platform, this controlled setup does not fully mirror the complexity of real-game scenarios. A more significant limitation is the absence of muscle activation data, which could provide a deeper understanding of the neuromuscular strategies involved in volleyball block jumping. Future research should aim to simulate more realistic game conditions, incorporating varied stimuli and muscle activation analysis to better understand motor strategies in block jumping [28,58]. 

  1. Kindest regards

Comments on the Quality of English Language:

The paper contains minor grammatical errors, including inconsistent use of tense, lack of commas in complex sentences, and occasional awkward phrasing, which slightly detract from the overall readability but do not significantly affect the clarity of the research findings

Response: Thank you for your helpful comments regarding the quality of the English language in our manuscript. We have made the necessary revisions to address the minor grammatical errors, inconsistent tense usage, and awkward phrasing you mentioned. Additionally, we have marked these changes for your convenience. We appreciate your attention to detail and believe these improvements enhance the overall readability of our paper. Thank you!

Thank you again for your constructive and valuable criticism, and the time spent analyzing this Manuscript. I reiterate our sincere gratitude to you for your help and patience. Thank you very much!

Round 2

Reviewer 1 Report

Comments and Suggestions for Authors

Thank you very much for your thorough and thoughtful review. I sincerely appreciate the time and effort you dedicated to providing such detailed and constructive feedback

Reviewer 4 Report

Comments and Suggestions for Authors

Dear authors,

Thank you for taking on the original feedback and making amendments to enhance the quality of the manuscript. 

Kindest regards